# Reliability and Validity of the Comprehensive Assessment of Psychopathic Personality—Self-Report—German Version (CAPP-SR-GV) in a German Non-Criminal Sample

**DOI:** 10.3390/bs14090827

**Published:** 2024-09-17

**Authors:** Denis Köhler, Josephine Alexandra Boegel-Driessen, Jan Josupeit, Sarah-Joelle Issa-Keller, Romina Müller, Johannes Stricker

**Affiliations:** 1Faculty for Social Sciences and Cultural Studies, University for Applied Sciences Düsseldorf, 40489 Düsseldorf, Germany; 2Department of Clinical Psychology, Institute of Experimental Psychology, Heinrich Heine University of Düsseldorf, 40225 Düsseldorf, Germany; 3Faculty of Health and Nursing, University of Applied Sciences Gesundheit/Health Bochum, 44801 Bochum, Germany; 4Ambulatorium Psychiatry and Psychotherapie Rüschlikon, 8803 Rüschlikon, Switzerland; sarah.joelle.issa.keller@gmail.com

**Keywords:** Comprehensive Assessment of Psychopathic Personality, CAPP-SR, psychopathy, TriPM, personality

## Abstract

The structure of psychopathy is conceptualized differently in various models and no consensus has yet been reached. This study aimed to further clarify psychopathy’s content and structure by validating the German Comprehensive Assessment of Psychopathic Personality—Self-Report (CAPP-SR). For this purpose, we used a sample of *n* = 458 participants. The CAPP’s six factors were replicated in an exploratory factor analysis. Further confirmatory analysis revealed recommendations for optimizing the CAPP. Thus, both an optimized version and the original CAPP-SR were examined. The construct validity of both versions was then examined using a different self-rating instrument for psychopathy, the Triarchic Psychopathy Measure. The implications of the findings and further research directions are discussed.

## 1. Introduction

“Manie sans délire”, a term used by Pinel in 1806 to characterize psychopathic personalities, was the first description of psychopathic personality disorder. Psychopathy is therefore one of the earliest known personality disorders and at the same time the most famous among them [1]. Nevertheless, there is still no unified definition of the disorder and its etiology has not yet been fully explained [2,3]. The descriptions that are currently most influential are those of the American psychiatrist Hervey Cleckley [4]. According to Cleckley [4], psychopathic personality manifest itself in various areas of life through inconsiderate and uninhibited behavior and is masked by an apparently feigned psychological normality. The psychopathy construct coined by Cleckley identified psychopathy as a pathology that has features of clinical value and within which the mere presence of antisocial or criminal behavior is insufficient for diagnosis [3]. Westen and Weinberger noted that almost every current investigation of psychopathy is based on Cleckley’s descriptions [5]. Cleckley’s characterizations also provided a large part of the basis for Hare’s model of psychopathy [6]. The assessment tools derived from Hare’s model, in particular the Psychopathy Checklist [7], are now seen as the gold standard for assessing psychopathy [5,8]. However, other researchers question Hare’s model and its factor-based structure [9,10]. As a result, new explanatory models have been proposed such as those reflected by the Triarchic Psychopathy Measure [11,12] and the Comprehensive Assessment of Psychopathic Personality [13,14]. These newer models have advanced the understanding of psychopathy. In particular, the CAPP questions the antisocial or socially deviant behavior in the psychopathy construct [15] and addresses the question of whether these behaviors result from psychopathy or are a direct component of the construct [16,17]. Although psychopathic traits are not necessarily associated with delinquency [18], there is evidence of the predictive value of psychopathic personality disorder and future violent behaviour [19]. Not least for this reason, the assessment of psychopathic personality disorder traits, e.g., with the CAPP, is particularly important for the forensic context [20]. However, a further development guideline stated that personality disorders describe personal rather than social or cultural deviations. Thus violations of norms, such as criminal or antisocial forms of behavior, are explicitly not a component of items assessing psychopathy. Based on the dynamic nature of psychopathic personality [15], the CAPP model is a concept map that aims to fully capture the psychopathic traits identified by Cooke et al. [14] and the characteristics on which it is based. The dynamics assumed for personality disorders allow for a dimensional mapping of the changes in the type and severity of symptoms over time [21]. A bottom-up approach [22] was employed to develop the construct, taking into account literature reviews, expert opinions, relevant current research results, and direct observation of behavior [9]. The resulting CAPP includes six domains underlying psychopathy: (1) Dominance, which describes the hostile, dominating approach to other people; (2) Behavioral, with behavior typified by lack of endurance, restlessness and aggressiveness; (3) Attachment, which is characterized by loose, non-binding relationships lacking empathy and caring; (4) Self, which includes self-centeredness and an unstable concept of self; (5) Emotional, which covers such features as a lack of emotional stability; and (6) Cognitive, which includes facets such as intolerance and lack of concentration. For a graphic representation of the model with all its 99 items see Cooke et al. [14].

The CAPP also includes symptoms whose connection to psychopathic personality is not clear or is the subject of disagreement [15,23]. As is evident from its name, comprehensiveness is described as a central strength of the CAPP [14]. Additionally, the CAPP authors focus on comprehensibility by using to natural language and employing the lexical approach to personality [24]. As a result, the CAPP was subsequently found to be easy to translate, with comparable structures even across different language families [25], and has already been translated into over 25 languages [15]. Meanwhile, a variety of third-party assessment instruments (CAPP-Institutional Rating Scale (CAPP-IRS) [14]; CAPP-Staff Rating Form (CAPP-SRF) [14] and self-report instruments (CAPP-Lexical Rating Scale (CAPP-LRS; [10,25]; CAPP-Self-Report; CAPP-SR [16]) exist. Although the self-report instruments have some disadvantages due to the characteristics underlying the psychopathic personality, such as pathological lying and weak introspection [25], they also have advantages due to economic aspects concerning assessing social desirability [20] as well as the representation of socially undesirable characteristics in self-report procedures, which makes the use of these effective complementary procedures in the forensic context appear justified [26].

Especially in the research context, well-studied and valid self-report instruments are of great importance to further explore and define the psychopathy construct. Psychopathic traits can also be found to a less pronounced but significant extent in the general population, for whom the Psychopathy Checklist—Revised (PCL-R) cannot be used without problems [27,28]. Psychopathy research is therefore highly relevant in social terms as well as in psychological and forensic. Therefore, the present project aims to validate the CAPP-SR in German in a sample from the general population. To date there have been no studies on the German version of the CAPP-SR. Previous international validation studies indicate good psychometric properties across different cultures and samples [29,30]. Furthermore, a largely acceptable to excellent internal consistency of the CAPP-SR was found in a sample of offenders and primarily female students, both for the total score and the individual domains [31]. Only the domain attachment showed unacceptable internal consistency in the offender population [31]. In terms of convergent validity, high positive correlations were found between the CAPP-SR total score and the TriPM total score for both the general population [15] and offenders [31]. Moderate to high positive correlations with the overall TriPM score were also found at the domain level [15]. In terms of incremental validity, the CAPP-SR even turns out to be superior to the CAPP-LRS [15]. Analysis results regarding convergent validity are considered particularly problematic for unstable self-concept and lack of enjoyment [15], so the relevance of these traits to the construct should be discussed and excluded as appropriate, subject to further study [31]. 

Based on previous validation studies, the factor structure of the German CAPP-SR was investigated to assess the construct validity of the CAPP model. We expected to replicate a six-factor structure, identify satisfactory internal consistencies for all subscales, and indicators of satisfactory convergent validity with the TriPM scales. 

Moreover, we hypothesized that, as hierarchical structural equation models (SEM), each of the six domains of the CAPP-SR would demonstrate good fit. As well as validity, assessment instruments also need to demonstrate good reliability [32]. 

Concerning the approach to test both, the whole model and the domains individually we followed the idea of Sellbom, Cooke and Shou [15], who reasoned this with the conceptual similarity of the domains “and therefore, potential problems with discriminant validity” (p. 881) on the one hand and due to the concern of statistical artefacts. 

We therefore investigated the reliability of psychopathy assessment instruments. First, it was hypothesized that the CAPP-SR would possess good reliability, both overall and for all subscales.

To examine the criterion validity the correlations of the TriPM and CAPP-SR were investigated and it was hypothesized that the results would replicate those of Sellbom et al. [15]. Thus, a strong positive correlation between the overall scores of the TriPM and the CAPP-SR was hypothesized. For the subscales, earlier findings suggested that all the subscales of the TriPM except Boldness correlate strongly with the CAPP-SR.

## 2. Method

### 2.1. Procedure

Data were collected over a period of three months using a questionnaire which was presented in paper-and-pencil form and online on the questionnaire platform SoSciSurvey (http://www.soscisurvey.de, accessed on 13 September 2024). This made it possible to carry out direct questioning at two universities and also to recruit more widely online. The meta-analysis of Dodou and de Winter [33] found no differences in the social desirability of the two test forms. Furthermore, Ray et al. [26], in their meta-analysis, even found a weak negative correlation in non-forensic samples between self-reported psychopathic properties and the tendency to give socially desirable answers (“faking good”). Subclinical research on psychopathy has considerable importance, for example concerning etiology, and Cooke [9] suggested that the CAPP-SR be examined in an explicitly non-forensic sample. All subjects took part voluntarily and gave their consent to the processing of their data. At the end of the survey, the participants had the option to enter a “codeword”. Every “codeword” from all surveys were collected (a procedure regularly done for researches and studies at the Heinrich Heine University) and were used to grant an intern remuneration for students.

### 2.2. Assessment Instruments

#### 2.2.1. CAPP-Self Report German Version

To assess psychopathy according to the CAPP model, the German translation of the CAPP-SR [16,34] was used. The German CAPP-SR includes 99 items in the six domains Attachment, Behavioral, Cognitive, Dominance, Emotional, and Self. The items (e.g., “I can be quite slick”) can be answered using a 4-point scale ranging from 1 (false) to 4 (true). Studies carried out up to now [15] have found the CAPP-SR domain scales to have acceptable internal consistency (*ω* = 0.61–0.89) and to show mostly good criterion validity with the CAPP-LRS (*r* > 0.3 excluding the CAPP symptoms D6, E1, S2, S5). Evidence of the content validity of the CAPP model has been provided by studies of prototypicality [35,36,37].

#### 2.2.2. TriPM 

The TriPM is used to assess psychopathy according to the triarchic model [38,39]. This scale consists of the three subscales Boldness, Meanness and Disinhibition. The 58 items are judged on a four-point scale (true, somewhat true, somewhat false, false). By way of example, one item is “I return insults”. The internal consistency was very good (*α* > 0.8) in different languages, both overall and for the subscales [40]. There was also high convergent validity with other psychopathy instruments [41].

### 2.3. Sample

In total, *n* = 458 complete data sets were obtained, after adjustment for *n* = 21 outliers (*M* ± 2**IQR* for each scale). The sample size thus exceeded the recommendation of *n* = 33*5 = 165 data sets [42] for confirmatory modelling with 33 exogenous variables. The sample size also exceeded the recommended minimum sample size of *n* ≥ 300 for stable test parameters in exploratory factor analysis (EFA) [43]. In comparison, a minor difference was found between the online (*n* = 428) versus the paper-pencil version (*n* = 30) for the CAPP-SR total score (t(456) = −2.024, *p* = 0.044).

The sample consisted of 72.9% females, 25.8% males, and 1.3% people with other genders. Participants’ ages ranged from 17 to 84 years; the average age was 25.41 years (*SD* = 7.60). Most participants (90.4%) were students; over two thirds (67.9%) were undergraduates. The most frequent subjects were social sciences (35.6%), engineering (17.9%), and psychology (13.8%). Almost one person in five was working (17%) (either full- or part-time).

### 2.4. Statistical Evaluation and Testing of Model Assumptions 

The data were analyzed using IBM SPSS Statistics 25 and AMOS 25. The sample was found to be “meritorious” [44], having a Kaiser–Meyer–Olkin value of 0.891 for the representation of factors. The same was found for the suitability of each of the indicators (symptoms) involved as the Measure of Sampling Adequacy (*MSA*) values were all considerably greater than 0.600 (except *MSA_E1_* = 0.772 where *MSA* > 0.800 is required). Prior findings by Sellbom et al. [15] suggested a non-normal multivariate distribution of the CAPP-SR. As expected, the Kolmogorov–Smirnov test and the Shapiro–Wilk test both confirmed the non-normal multivariate distribution (*K-S*(458) = 0.051, *p* = 0.006, *S-W*(458) = 0.984, *p* < 0.001); the distribution was clearly skewed to the right (skew = 0.314) and platykurtic (kurtosis = −0.533). The consequences were therefore: (1) for EFA, robust principal axis factor analysis with direct oblimin rotation; this approach allows and shows correlations between factors and—where factors are not correlated—gives almost identical results to orthogonal rotations [45]; (2) a low bias for the estimated factor loadings using maximum likelihood estimation [46] and higher Type1 error for rejection of the *χ*^2^-Test [47], which was reduced because the condition described by Savalei [48] for a robust ML-estimate in the case of a non-normal distribution (e.g., complete data) was fulfilled; (3) the additional inclusion of the fit indices Comparative Fit Index (*CFI*) and Root Mean Square Error of Approximation (*RMSEA*), taking *CFI* > 0.95 and *RMSEA* < 0.05 to be good indicators of fit [49]. In the CAPP-SR as a whole, each SEM was modeled in the third order or in the individual domains in the second order. Potential modifications were carried out post hoc if the fit was not sufficiently good by excluding items that may have been too broadly defined [15,23] depending on their prototypicality, item formulations, content and descriptive values, correlation of covarying errors within a symptom and combination of symptoms. Modification Index (*M.I*), factor loadings, mean, skew, kurtosis, and item difficulty served as statistical indicators. Each modification was carried out according to the Single Step Modification Approach [50] so that only one change was carried out at a time and the changed model was adopted only if global goodness of fit was improved and the parameters were plausible.

Cronbach’s Alpha for internal consistency was used as a reliability coefficient and was calculated before correlation testing. The minimum criterion of *α* higher than 0.700 [32] was fulfilled for the whole scales of the unmodified CAPP-SR (*α* = 0.928) and the TriPM (*α* = 0.815). The subscales also showed lower reliabilities, although these remained above the minimum of 0.500 for psychological constructs in the early stages of research [51]. Finally, Pearson product-moment correlations were calculated. Pearson’s coefficient *r* was used because it is robust concerning violations of its statistical assumptions of interval scaling of the data and bivariate normal distribution of both traits [52].

## 3. Results

### 3.1. Factor Structure of the CAPP-SR 

First of all, the general structure of the CAPP-SR was examined using EFA. This involved investigating the 33 symptoms using principal axis factor analysis as an extraction method and direct oblimin rotation for correlated factors. 

The scree plot analysis to find the ideal number of factors for large samples (*n* > 300) [32] resulted in six factors. In all, the six factors explained 57.81% of the total variance. The eigenvalues were 8.039, 4.370, 2.176, 1.835, 1.416, 1.241. The regression weights of the pattern matrix are shown in Table 1. In 29 of the 33 cases, the highest semipartial regression weights were also equivalent to the highest correlations of the items with the factors of the structure matrix. Some of the domains showed a tendency to belong to a shared factor (Dominance to 1, Attachment to 6, Behavioral to 3); some showed great variability within the shared factor (Cognitive, Emotional, Self). Moderate correlations (*r* ≥ 0.300) were found between individual extracted factors (*r*_13_ = 0.333, *r*_15_ = 0.336, *r*_16_ = 0.397, and *r*_35_ = 0.392).

Finally, the results show a six-factorial solution, but not a simple structure with respect to the postulated model. There are numerous many cross-loadings and many items with small loadings. The whole model was examined using higher-order confirmatory factor analysis (CFA) in a reflective model with all 33 symptoms allocated to the six domains. In line with the EPA’s findings the goodness of fit was not acceptable (*χ*^2^(489) = 3513.136, *p* < 0.001, *CFI* = 0.499, *RMSEA* = 0.116).

### 3.2. Confirmatory Testing of the Domain Structures

The structure of the individual domains was also tested using the relevant SEM. None of the models achieved acceptable goodness of fit. See Table 2 for all quality criteria of the SEMs.

### 3.3. Modification of the Domain Structures of the CAPP-SR

Because none of the original CAPP-SR models showed adequate goodness of fit, the SEMs of the individual domains were modified.

In the domain Dominance, Item 10 (D6) was deleted because it showed a strong correlation with Item 12 (*M.I.* = 30.048, *Par Change* = 0.170), although the two were not correlated in terms of content. In addition, D6 as a whole showed moderate to low prototypicality [35] and was also removed in previous CFA studies [36]. All other modifications followed this pattern. The final modified Dominance model was also adjusted to exclude Item 1 as well as the symptom Deceitful (D3) because these items covary strongly with other symptoms and there were difficulties of formulation that were apparent in skew, kurtosis and item difficulty respectively. In addition, the errors e_12_-e_20_, e_96_-e_80_, e_53_-e_05_ and e_37_-e_05_ were correlated.

Within the Cognitive domain, the symptom Lacks concentration (C2) was removed because of strong criticism of the content, non-prototypicality and similarity to control items [35,36,37]. 

In the domain Attachment, the errors e_33_-e_81_ (*M.I.* = 9.569, *Par Change* = 0.072), e_03_-e_72_ (*M.I.* = 5.555, *Par Change* = 0.036), and e_48_-e_L_ (*M.I.* = 4.905, *Par Change* = −0.053) were correlated.

In the domain Behavior, the errors e_35_-e_70_ (*M.I.* = 95.203, *Par Change* = 0.094) and e_14_-e_65_ (*M.I.* = 5.288, *Par Change* = 0.049) were correlated. In addition, strong connections were found between the symptoms Disruptive (B5) and Aggressive (B6), *M.I.* = 32.221 (*Par Change* = 0.116) and also between their items. Their content can be summarized as “Powerful/Controlling”. The errors e_76_-e_09_ and e_49_-e_27_ now belonged to one symptom and were correlated. Finally, the whole symptom Lacks perseverance (B1) was removed because it had moderate to low prototypicality depending on the sample. In the prison inmate sample, it was even equal to the control items [35,36]. Item 17 was removed because it was strongly confounded (e.g., with e_AUS_: *M.I.* = 49.586, *Par Change* = 0.130, with e_R_: *M.I.* = 11.778, *Par Change* = 0.087).

The Emotional domain was adjusted to remove the symptoms Lacks remorse (E5) and Lacks anxiety (E1). These two symptoms gave the remaining symptoms opposing factor loadings and clearly showed many difficulties in their items as a result of long formulations, multiple conditions, insight into their own errors or the subjunctive. These difficulties were clearly apparent in properties such as skew, *M.I*. and item difficulty. E1 (Lacks anxiety) also showed moderate to low prototypicality [35]. Within the symptom Lacks emotional stability (E4), which was prototypical for other patients as well, Item 88—its strongest item—was removed.

In the domain Self, the symptom Unstable self-concept (S7) was removed because of theoretical confounding with borderline personality disorder and also because of low prototypicality, sometimes being judged the same as control items [35,36,37]. Additionally, the weakest item of each symptom with regard to comprehensibility, symptom specificity and referring statistical values was eliminated: Item 60 (S1), 22 (S2), 34 (S3), 38 (S4), 56 (S5) and Item 40 (S6). The symptom Self-justifying (S6) was strongly correlated with the symptom Sense of entitlement (S4) (*M.I*. = 15.219, *Par Change* = −0.057), also at item level (e.g., Item 68 with S4: *M.I*. = 25.301, *Par Change* = 0.088), so that Self-justifying (S6) was removed.

The goodness of fit and fit indices of all modified models are shown in Table 3.

### 3.4. Reliabilities of the CAPP-SR and the TriPM

Table 4 shows the reliability coefficient Cronbach’s Alpha for the TriPM and for the CAPP-SR in unmodified and modified form.

Both psychopathy assessment instruments showed good reliabilities of *α* ≥ 0.700 [32] in the whole scale (*α_CAPP-SR_*′ = 0.914, *α_TriPM_* = 0.815), and also in all subscales (*α_Emotional_*′ = 0.700 to *α_Behavioral_*′ = 0.781, *α_Meanness_* = 0.787, *α_Boldness_* = 0.815). One exception was the original CAPP-SR domain Emotion, which had, prior to the modification, a rather low reliability (*α_E_* = 0.556).

### 3.5. Correlations between the CAPP-SR and the TriPM

Forty-four out of 49 correlations of the TriPM with the modified CAPP-SR (*r_M_*) were the same as those with the original version (*r_O_*). All correlations are shown in Table 5. The majority of the correlations between the TriPM and both versions of the CAPP-SR were positive, both for the overall values and for the subscales, and significant at the level of *p* < 0.001.

The weak correlation between the modified Emotional domain and the TriPM total score (*r_M_* = 0.132, *p* = 0.005) and the negative correlations of the TriPM subscale Boldness are exceptions. The single nonsignificant correlation between the modified CAPP-SR domains Self and Emotion (*r_M_* = 0.001, *p* = 0.988) is also striking.

## 4. Discussion

Finally, the results can be summarized as follows. The study was first intended to examine reliability and was able to determine overall satisfactory to good values. The factor structure of the CAPP-SR was confirmed through an exploratory factor analysis. Surprisingly, the results of the confirmatory factor analysis were not as clear as expected. However, better results could be achieved by testing a modified model. The criterion validity could be confirmed by a comparison with the TriPM. These results are in line with previous research. The results are discussed in more detail below.

In the EFA, examination of the CAPP-SR structure provided evidence in support of an underlying six-factor structure. Due to the lack of a single structure and the numerous incorrect and low item loadings, the factor structure assumed by Cooke et al. [14] could only be replicated to a limited extent. The symptoms of Dominance, Attachment and Behavior were found to load mainly on one factor each, although their factors showed moderate correlations with each other. However, more precise analysis of the allocation of symptoms to factors permits other types of hypotheses that differ from those envisaged in the CAPP model. Also, the CAPP-SR overall did not show sufficient goodness of fit.

The models of the six CAPP-SR domains likewise failed to show sufficient goodness of fit. This finding is in line with the results of the EFA and agrees with the basic notion that the CAPP may be too broadly based [15,23].

For all domain models, it was possible to achieve good model fit through modification. The special feature of this CFA study lies in the stepwise modification at the item level. This takes into account the fact that only single-item formulations can cause problems (as opposed to adjective-based CAPP procedures) [15], this is especially likely where translations are used. It was thus possible to achieve an adequate improvement of model fit in some cases (e.g., by deleting Item 10 instead of D6, Garrulous). Overall, more information was retained and it became possible to make modifications closer to the original concept.

This approach achieved good model fit for the domain Self, which has seven symptoms and is the broadest of the domains, while symptom deletions made the model fit worse. 

CFA studies carried out up to now [36] were replicated for some domains (e.g., Dominance, Cognition) but lead to considerable worsening of the model in other domains. In the present study, deletion of Lacks pleasure (E2) does not lead to replication of the improvement in CFA fit reported in Hoff et al. [36]. In its original modeling, the Emotional domain already had the special characteristic that Lacks anxiety (E1) and Lacks remorse (E5) demonstrated opposite factor loadings to the remaining symptoms. A possible reason for this is that psychopaths do not lack feelings such as joy and anxiety as such, but instead there are deficits in the emotional repertoire relating to others [53,54,55]. This hypothesis is supported by the relationship found in the EFA between the symptoms Lacks pleasure (E2) and Lacks emotional depth (E3), though when these were merged into one symptom in the CFA, there was no resulting improvement in CFA fit and the merged symptom had to be discarded.

Often, the prototypicality studies also yielded striking statistical parameters, e.g., a floor effect with large skew. The same applied to difficult item formulations. The Emotional domain, in particular, reflected the problem self-assessment instruments have with putting all their theoretical assumptions in practical terms. In the items, this is reflected in such things as several conditions (e.g., “With the exception of anger, I have never been one to express emotions”) or the inclusion of other people’s judgements (e.g., “Others seem to think that I am not emotionally expressive”).

It was also striking that some item errors were strongly covariant as a result of using three items as synonyms for one symptom [14]. Throughout all domain modifications, there were correlations of item errors within one symptom. For the domain Attachment, this step was already sufficient to fulfill the criteria for good fit.

One last type of modification concerned the combining of symptoms with similar content and of symptoms showing statistical correlations with each other. In some cases (e.g., Disruptive, B5, and Aggressive, B6), this created a new symptom (Powerful-controlling) that improved the model, but in other cases, it was necessary to eliminate a symptom (e.g., Sense of entitlement, S4, and Self-justifying, S6).

Overall, the modifications uphold the importance of prototypicality studies and previous CFA studies as good indicators for model respecification. In particular, the prototypicality studies delivered important findings with regard to differentiation from other psychological disorders such as borderline personality disorder (e.g., relating to Lacks emotional stability, E4, and Unstable self-concept, S7).

The overall model produced by combining the modified domain models did not achieve good fit and gave only a slight improvement compared to the original model. Items may have been deleted at the domain level that can only be allocated to another domain in the overall model [23]. It would therefore be useful to analyze the original overall model including all results up to now. It is possible that the CAPP model is too broad, which often leads to difficulties when assessing model fit. However, conceptually, the broad conceptualization of psychopathy in the CAPP model is an advantage over alternative models because of its clinical benefits. A disadvantage of such broad-based personality models is the need for very large samples and specially tailored statistical analysis procedures. These points should be taken into account in future studies.

Due to the removal of items or symptoms on the basis of statistical parameters (CFA), it should be noted that this procedure is logically opposed to the development of the CAPP, which was developed with the help of lexical and prototypical analyses. In principle, however, such a cross-check is also necessary to test the validity of a personality model. By removing items from the statistical analyses, significant information from the psychopathy construct could, of course, be lost. However, it should be noted that individual items or trait-descriptive adjectives were only removed at the lowest hierarchical level. This approach removes information from the model or the domains, but by no means fundamental information. Nor does it call into question the factor structure of the CAPP per se. Rather, future research on the CAPP should critically examine the postulated triangulation of three trait-descriptive adjectives (one item each), each of which depicts one symptom, and revise or better operationalize it if necessary.

The reliability of the modified CAPP-SR was good throughout. The Emotional domain, in particular, achieved the threshold for good reliability, *α* = 0.700, only as a result of modification. The TriPM also achieved good reliabilities above the minimum level, both overall and for all subscales [32]. 

The next examination found predominantly positive correlations between the psychopathy assessment instruments and thus provided evidence that the CAPP-SR possessed good criterion validity. The overall scores of both CAPP-SR versions showed strong positive correlations with the TriPM overall score (*r_O_* = 0.680, *r_M_* = 0.638, *p* < 0.001). It was found that TriPM Boldness did not correlate significantly with the subscales of the CAPP versions, thus replicating the findings of Sellbom et al. [15]. Only for Dominance and Self did both CAPP-SR versions show low to moderate correlations with TriPM Boldness. 

The nonsignificant correlation of the Emotional domain with the domain Self in the modified CAPP-SR is a peculiarity. It is possible that the modification eliminated previous potential overlap between the two domains. Further research needs to be carried out into the content and interactions of the two domains.

Overall, the results are mostly in line with the prior findings of Sellbom et al. (2020) [31] and support the construct validity of the CAPP Model. Nevertheless, future studies should pay attention to a higher heterogeneity of the sample, especially with regard to socio-demographic variables (e.g., age, level of education, cultural background).

## 5. Limitations and Research Prospects

The investigation of a non-forensic sample offers opportunities [9] but naturally restricts the psychopathy range [15]. One potential area for future research is to compare the different factor structures of a subclinical sample with those of another (e.g., forensic) sample. 

The results of the present study are largely consistent with the existing CAPP model. In the translation process, a back translation with native speaker checking was carried out to ensure maximum linguistic equivalence. Nonetheless, due to linguistic subtleties deviations could possibly have occurred due to a lack of content fit of the original English items to the German translations. In this respect, further research should examine whether a better model fit can be achieved by means of a modified German translation of the items. Because of the considerable overrepresentation of students (90.4%), further validation is to recommended in a heterogeneous sample that is more representative of the general population. The same goes for the overrepresentation of women (72.9%), as Kreis and Cooke [37] had already found significant gender differences in the CAPP. As well as the replication of previous results, further research on reliability and validity could include areas such as the examination of the retest reliability and the parallel test reliability in the context of behavior-based psychopathy assessments [26].

A further and important limitation of the study is that no measurement invariance was considered. Kreis and Cooke already pointed out possible gender differences in 2011, but could not find any evidence for fundamental prototypical differences. The current study by Spohrmann, Mokros and Schneider [55] investigated the measurement invariance of different psychopathy models and instruments. They were mainly able to demonstrate higher mean scores in men compared to women. However, there were hardly any gender differences in the factor structure of the models/instruments investigated. In future studies on the CAPP-SR, more emphasis should be placed on measurement invariance by analyzing larger samples with a similar gender distribution.

In sum, the German translation of the self-assessment version of the CAPP has proven to be a reliable and valid assessment tool; it was considerably improved by a few modifications. Previous research on this new conceptualization of psychopathy has been promising and should be replicated and enlarged upon in further research. 

## Figures and Tables

**Table 1 behavsci-14-00827-t001:** Principal axis factor analysis of the CAPP-SR symptoms.

		Factors					
		1	2	3	4	5	6
Dominance	Antagonistic (D1)	0.000	**0.629 ***	0.038	0.006	0.358	0.217
	Domineering (D2)	0.175	0.021	−0.237	**0.539 ***	0.179	0.037
	Deceitful (D3)	**0.840 ***	−0.028	0.002	−0.083	−0.131	0.034
	Manipulative (D4)	**0.634 ***	0.030	−0.067	0.098	0.075	0.076
	Insincere (D5)	**0.552 ***	−0.091	0.088	0.081	0.080	−0.034
	Garrulous (D6)	**0.349 ***	−0.050	0.113	0.295	0.076	−0.132
Cognitive	Suspicious (C1)	0.191	−0.140	−0.079	0.070	0.305	**0.317 ***
	Lacks concentration (C2)	−0.108	−0.123	**0.747 ***	0.076	0.067	0.106
	Intolerant (C3)	0.220	−0.084	0.017	**0.439 ***	0.089	0.364
	Inflexible (C4)	−0.049	0.011	0.032	0.116	**0.547 ***	0.134
	Lacks planfulness (C5)	0.121	0.155	**0.654 ***	−0.236	0.054	−0.057
Attachment.	Detached (A1)	0.014	−0.047	0.052	−0.035	0.052	**0.668 ***
	Uncommitted (A2)	0.266 *	0.067	0.202	0.017	−0.067	**0.317**
	Unempathic (A3)	0.381 *	0.200	0.044	0.009	-0.100	**0.427**
	Uncaring (A4)	0.176	0.229	0.016	−0.087	0.220	**0.368 ***
Behavior	Lacks perseverance (B1)	−0.061	−0.166	**0.810 ***	0.047	−0.061	0.173
	Unreliable (B2)	0.173	−0.101	**0.579 ***	−0.107	0.000	0.113
	Reckless (B3)	0.178	0.281	**0.503 ***	−0.064	0.250	−0.177
	Restless (B4)	0.030	0.086	**0.381 ***	0.157	0.266	−0.165
	Disruptive (B5)	0.097	0.193	0.198	0.093	**0.358 ***	0.051
	Aggressive (B6)	0.270 *	0.303	−0.053	0.067	**0.361**	0.076
Emotion	Lacks anxiety (E1)	−0.014	**0.784 ***	−0.095	0.141	−0.103	−0.005
	Lacks pleasure (E2)	−0.060	−0.394	−0.036	−0.192	0.327	**0.431 ***
	Lacks emotional depth (E3)	−0.019	0.049	0.068	0.047	0.012	**0.698 ***
	Lacks emotional stability (E4)	−0.049	−0.081	0.195	0.005	**0.650 ***	−0.019
	Lacks remorse (E5)	0.242 *	**0.339**	0.047	0.098	−0.047	0.281
Self	Self-centered (S1)	0.087	−0.046	0.318	0.368	−0.032	**0.382 ***
	Self-aggrandizing (S2)	−0.037	0.207	−0.010	**0.707 ***	−0.205	−0.068
	Sense of uniqueness (S3)	0.004	0.172	−0.054	**0.639 ***	0.054	0.028
	Sense of entitlement (S4)	0.068	−0.060	0.064	**0.591 ***	0.285	0.010
	Sense of invulnerability (S5)	−0.120	**0.565 ***	−0.057	0.361	−0.100	−0.079
	Self-justifying (S6)	0.163	−0.153	0.105	0.031	**0.418 ***	−0.011
	Unstable self-concept (S7)	0.048	**−0.381 ***	0.243	−0.022	0.227	0.104

Note: Pattern matrix of the principal axis factor analysis, oblimin rotation method with Kaiser normalization. The rotation converged in 36 iterations. The highest semipartial standardized regression weights are printed in bold type. * = this element represents the highest correlation between an item and a factor in the structure matrix. *n* = 458.

**Table 2 behavsci-14-00827-t002:** Global quality criteria of the CAPP-SR model as a whole and of the individual domains.

	*χ* ^2^	*Df*	*χ*^2^/*df*	*CFI*	*RMSEA*
Dominance	422.238 *	129	3.273	0.843	0.071
Cognitive	227.697 *	85	2.679	0.925	0.061
Attachment	122.384 *	50	2.448	0.911	0.056
Behavioral	505.629 *	129	3.920	0.834	0.080
Emotional	351.208 *	85	4.132	0.823	0.083
Self	687.939 *	182	3.780	0.770	0.078
Total	3513.136 *	489	7.184	0.499	0.116

Note: *χ*^2^ = Chi^2^, *df* = degrees of freedom, *CFI* = Comparative Fit Index, *RMSEA* = Root Mean Square Error of Approximation, * = *p* < 0.001. *n* = 458.

**Table 3 behavsci-14-00827-t003:** Global quality criteria of the modified CAPP-SR model as a whole and of the individual domains.

	*χ* ^2^	*Df*	*χ* ^2^ */df*	*CFI*	*RMSEA*
Dominance	109.243 ***	56	1.951	0.958	0.046
Cognitive	92.888 ***	50	1.858	0.962	0.043
Attachment	87.862 ***	47	1.869	0.950	0.044
Behavioral	142.841 ***	70	2.041	0.956	0.048
Emotional	32.359 *	17	1.903	0.982	0.044
Self	64.408 *	30	2.147	0.957	0.050
Total	1658.183 *	269	6.164	0.622	0.106

Note: = modified version, *χ*^2^ = *Chi*^2^, *df* = degrees of freedom, *CFI* = Comparative Fit Index, *RMSEA* = Root Mean Square Error of Approximation, * = *p* < 0.05, *** = *p* < 0.001. *n* = 458.

**Table 4 behavsci-14-00827-t004:** Reliabilities and descriptive parameters of the unmodified and modified CAPP-SR and TriPM.

		Items	Range	*M*	*SD*	*A*
CAPP-SR’	Dominance Cognitive	18 (13) 15 (12)	1–4 1–4	1.940 (1.956) 1.920 (1.837)	0.383 (0.416) 0.399 (0.386)	**0.808 (0.770)** **0.789 (0.724)**
	Attachment	12 (12)	1–4	1.543 (1.543)	0.375 (0.375)	**0.744 (0.744)**
	Behavioral	18 (14)	1–4	1.740 (1.719)	0.384 (0.425)	**0.821 (0.781)**
	Emotional	15 (8)	1–4	1.889 (1.804)	0.297 (0.450)	0.556 **(0.700)**
	Self	21 (10)	1–4	2.080 (1.937)	0.332 (0.454)	**0.746 (0.757)**
	Total	99 (69)	1–4	1.852 (1.799)	0.278 (0.296)	**0.928 (0.914)**
TriPM	Boldness	19	0–3	1.547	0.413	**0.815**
	Meanness	19	0–3	0.488	0.312	**0.787**
	Disinhibition	20	0–3	0.647	0.322	**0.793**
	Total	58	0–3	0.894	0.224	**0.815**

Note: CAPP-SR = Comprehensive Assessment of Psychopathic Personality Self-Report, TriPM = Triarchic Psychopathy Measure, *M* = mean, *SD* = standard deviation, *α* = Cronbach’s Alpha. Values of Cronbach’s Alpha higher than 0.700 are printed in bold type. The values of the modified CAPP-SR are given in brackets. *n* = 458.

**Table 5 behavsci-14-00827-t005:** Intercorrelation matrix of the unmodified and modified CAPP-SR and the TriPM.

		1	2	3	4	5	6	7	8	9	10
CAPP-SR	Dominance (1)	-									
	Cognitive (2)	**0.495** (**0.574**)	-								
	Attachment (3)	**0.526** (**0.514**)	**0.518** (**0.542**)	-							
	Behavioral (4)	**0.487** (**0.484**)	**0.745** (**0.600**)	**0.426** (**0.378**)	-						
	Emotional (5)	**0.471** (**0.321**)	**0.569** (**0.579**)	**0.642** (**0.512**)	**0.479** (**0.358**)	-					
	Self (6)	**0.563** (**0.495**)	**0.457** (**0.321**)	**0.325** (**0.255**)	**0.448** (**0.263**)	**0.432** (0.001)	-				
	Total (7)	**0.774** (**0.790**)	**0.833** (**0.838**)	**0.746** (**0.734**)	**0.790** (**0.720**)	**0.763** (**0.648**)	**0.691** (**0.558**)	-			
TriPM	Boldness (8)	**0.319** **(0.253)**	**−0.163** (−0.085)	−0.007 (−0.007)	−0.006 (0.000)	*0.110* (**−0.365**)	**0.268** (**0.459**)	0.104 (0.064)	-		
	Meanness (9)	**0.647** **(0.625)**	**0.516** **(0.540)**	**0.661** **(0.661)**	**0.499** **(0.445)**	**0.559** (**0.360**)	**0.398** (**0.377**)	**0.714** (**0.697**)	**0.187**	-	
	Disinhibition (10)	**0.353** **(0.385)**	**0.621** **(0.550)**	**0.328** **(0.328)**	**0.680** **(0.663)**	**0.366** (**0.394**)	**0.330** (*0.133*)	**0.591** (**0.571**)	**−0.164**	**0.364**	-
	Total (11)	**0.667** **(0.632)**	**0.438** **(0.463)**	**0.461** **(0.461)**	**0.554** **(0.525)**	**0.504** (*0.132*)	**0.508** (**0.522**)	**0.680** (**0.638**)	**0.625**	**0.755**	**0.549**

Note: CAPP-SR = Comprehensive Assessment of Psychopathic Personality Self-Report, TriPM = Triarchic Psychopathy Measure. Significant correlations with *p* < 0.001 are printed in bold type, significant correlations with *p* ≤ 0.050 are printed in italic type. The correlations of the modified CAPP-SR are given in brackets. *n* = 458.

## Data Availability

The research data can be made available on request.

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
