# Peer review of "Reliability and Validity of the Comprehensive Assessment of Psychopathic Personality—Self-Report—German Version (CAPP-SR-GV) in a German Non-Criminal Sample"

_behavsci, 2024, doi:10.3390/bs14090827_

Round 1

Reviewer 1 Report

Comments and Suggestions for Authors

In the introduction, the theoretical basis of the paper is clearly presented, and it ends with the hypotheses.

In the sentence: "Concerning the approach to test both, the whole model and the domains individually we followed the idea of ​​Sellbom, Cooke and Shou [16], wo reasoned this with the" is tipfeler.

For CAP - SR, I consider it inadequate to say that the answers are given on a Likert scale because attitudes are not measured. It is less appropriate to say that a Likert-type scale is used, because it is not expressed how much the participants agree with the content of the statement, but how true or false it is for them, so I would recommend stating that the answers are given on a four-point scale. All the above also applies to TriPM.

The authors present the results of the analysis that checks the appropriateness of the German translation with the original and the criterion validity, and briefly comment on the same in the discussion. In limitations, the authors validly state the challenge of using a non-clinical sample that will have less variation in psychopathy indicators.

Author Response

Dear reviewer, thank you very much for the constructive and positive feedback. We have incorporated all suggestions for improvement.

Comment1: "In the sentence: "Concerning the approach to test both, the whole model and the domains individually we followed the idea of Sellbom, Cooke and Shou [16], wo reasoned this with the" is tipfeler."

1) We have corrected this typo". Thank you for reading carefully.

Comment 2: "For CAP - SR, I consider it inadequate to say that the answers are given on a Likert scale because attitudes are not measured. It is less appropriate to say that a Likert-type scale is used, because it is not expressed how much the participants agree with the content of the statement, but how true or false it is for them, so I would recommend stating that the answers are given on a four-point scale. All the above also applies to TriPM."

2) We have changed the term Likert-scale to "answers are given on a four-point scale" in all places in the manuscript.

Reviewer 2 Report

Comments and Suggestions for Authors

This is a well-written paper, but I have several major and minor issues with it. My main issue is that the paper seems unfinished in providing evidence for the usefulness of the CAPP-SR in the German context. The paper factor analyzes the 33 items of the CAPP-SR, makes some suggestions about how to improve the psychometrics of the CAPP-SR, and then correlates the dimensions of the measure with the TriPM. The paper does not contain any additional evidence such as construct validity, concurrent validity, peer-ratings, etc., which are typical in papers like this. I think that the Editor has to make a tough choice whether this is enough to warrant publication in a Q2 journal.

Here, I list my other concerns in the order that I found them in the paper:

*Page 3 has a minor typo: “wo” instead of “who.”

*Some of the students were given remuneration for their participation. How did you ensure anonymity for these students?

*I suggest providing sample items for each domain/dimension for both measures (CAPP-SR and TriPM), not just for the overall scale.

*Has the TriPM been adapted and validated in German? This is a crucial issue as the correlations with TriPM provide the only source of validity for the CAPP-SR. I found only an unpublished work in the reference list, which is not encouraging.

*Were there differences between the findings in the online versus the paper-pencil version?

*Table 1 and Table 5 are hard to read; I suggest you modify them a bit.

*You claim that the EFA provided evidence in support of an underlying six-factor structure as proposed by Cook et al. As Table 1 is hard to read, I’m not sure I’m right about this, but to me, it seems that the factor structure went significantly differently than it should be. I see many cross-loading items on the table and many items with small loadings. To me, it seems that neither the EFA nor the CFA supported the original factor structure of the CAPP-SR, which is not a problem in itself but should be addressed in the paper.

Author Response

Comments and Suggestions for Authors

Comment: This is a well-written paper, but I have several major and minor issues with it. My main issue is that the paper seems unfinished in providing evidence for the usefulness of the CAPP-SR in the German context. The paper factor analyzes the 33 items of the CAPP-SR, makes some suggestions about how to improve the psychometrics of the CAPP-SR, and then correlates the dimensions of the measure with the TriPM. The paper does not contain any additional evidence such as construct validity, concurrent validity, peer-ratings, etc., which are typical in papers like this. I think that the Editor has to make a tough choice whether this is enough to warrant publication in a Q2 journal.

Here, I list my other concerns in the order that I found them in the paper:

  1. Comment: *Page 3 has a minor typo: “wo” instead of “who.”

Response: We have corrected this typo. Thank you for reading carefully.

  1. Comment: *Some of the students were given remuneration for their participation. How did you ensure anonymity for these students?

Response: Thank you for this note. We have included the following in the manuscript to clarify this.

“ At the end of the survey, the participants had the option to enter a "codeword". Every "codeword" from all surveys were collected (a procedure regularly used for research projects at the Heinrich Heine University Düsseldorf) and were used to grant an intern remuneration for students.”

  1. Comment *I suggest providing sample items for each domain/dimension for both measures (CAPP-SR and TriPM), not just for the overall scale.

Response: Thank you for this suggestion. We did not do that for a number of reasons:

- For economic reasons, we have tailored the paper stringently and briefly to the really statistically and methodologically significant aspects.

- The items translated into German would then have had to be listed. This does not seem to be expedient for non-German-speaking readers and the article is not about the concrete translation process.

- The copyright of the items or the instruments seems to be particularly relevant to us. Therefore, we have decided not to disclose this information in the article, also for legal reasons.

- Nevertheless, we have taken up the aspect by pointing out the corresponding sources with footnotes, where interested readers can receive the information about the items.

With this solution, we have tried to take into account the reviewer’s comments.

  1. Comment: *Has the TriPM been adapted and validated in German? This is a crucial issue as the correlations with TriPM provide the only source of validity for the CAPP-SR. I found only an unpublished work in the reference list, which is not encouraging.

Response: The German version of the TriPM has been validated and the related publication is ,according to the authors, in preparation. In addition, this German version has been used in several published studies, including validity studies.

Refs:

Kelley, S. E., van Dongen, J. D. M., Donnellan, M. B., Edens, J. F., Eisenbarth, H., Fossati, A., Howner, K., Somma, A., & Sörman, K. (2018). Examination of the Triarchic Assessment Procedure for Inconsistent Responding in six non-English language samples. Psychological Assessment, 30(5), 610–620. https://doi.org/10.1037/pas0000485

Hofmann, M. J., Mokros, A., & Schneider, S. (2023). The joy of being frightened: Fear experience in psychopathy. Journal of Personality, 00, 1–21. https://doi.org/10.1111/jopy.12890

Burghart, M., Sahm, A. H. J., Schmidt, S., Bulla, J., & Mier, D. (2024). Understanding empathy deficits and emotion dysregulation in psychopathy: The mediating role of alexithymia. PLOS ONE, 19(5), e0301085. https://doi.org/10.1371/journal.pone.0301085

Eisenbarth, H., Hart, C. M., Zechmeister, J., Kudielka, B. M., & Wüst, S. (2021). Exploring the differential contribution of boldness, meanness, and disinhibition to explain externalising and internalising behaviours across genders. Current Psychology. https://doi.org/10.1007/s12144-021-02134-3

  1. Comment: *Were there differences between the findings in the online versus the paper-pencil version?

Response: Thank you for this important note. We have included the following in the manuscript to take this aspect into account.

“In comparison, minor difference was found between the online versus the paper-pencil version (N=30) for the CAPP-SR total score (t(456) = -2.024, p = .044).”

  1. Comment: *Table 1 and Table 5 are hard to read; I suggest you modify them a bit.

Tables 1 and 5 are standard tables for representing EFA and intercorrelations, so we have designed them accordingly and would like to leave them as they are. This form is also used in the same way in comparable papers.

  1. Comment: *You claim that the EFA provided evidence in support of an underlying six-factor structure as proposed by Cook et al. As Table 1 is hard to read, I’m not sure I’m right about this, but to me, it seems that the factor structure went significantly differently than it should be. I see many cross-loading items on the table and many items with small loadings. To me, it seems that neither the EFA nor the CFA supported the original factor structure of the CAPP-SR, which is not a problem in itself but should be addressed in the paper.

Response: Thank you very much for these notes. We should have highlighted these aspects more clearly. Accordingly, we have listed the aspects more explicitly in the results section and in the discussion.

  • “Finally, the results show a six factorial solution, but it does not show a simple structure with respect to the postulated model. There are numerous cross-loadings and many items with small loadings. In line with the EPA's findings, the whole model was examined using higher-order confirmatory factor analysis (CFA) in a reflective model with all 33 symptoms allocated to the six domains. The goodness of fit was not acceptable (χ²(489) = 3513.136, p < .001, CFI = .499, RMSEA = .116).”
  • “In the EFA, examination of the CAPP-SR structure provided evidence in support of an underlying six-factor structure. Due to the lack of a single structure and the numerous incorrect and low item loadings, the factor structure assumed by Cooke et al [15] could only be replicated to a limited extent. The symptoms of Dominance, Attachment and Behavior were found to load mainly on one factor each, although their factors showed moderate correlations with each other. However, more precise analysis of the allocation of symptoms to factors permits other types of hypotheses that differ from those envisaged in the CAPP model. Also, the CAPP-SR overall did not show sufficient goodness of fit."
  • "The models of the six CAPP-SR domains likewise failed to show sufficient goodness of fit. This finding is in line with the results of the EFA and This dovetails with the basic notion that the CAPP may be too broadly based [16,23].”

Round 2

Reviewer 1 Report

Comments and Suggestions for Authors

Thank you for adopting the remarks. 

Author Response

Comment 1:

The manuscript is relevant and is studied in a non-criminal population, which is appropriate given the broad application of psychopathy measurement tools in clinical and forensic settings, as well as in the general population. The literature review is well-done and provides a good overview of the CAPP-SR. The authors used both EFA/CFA to examine the factor structure.

However, research on measurement invariance is missing. This is an important issue and cannot be easily captured/reviewed due to the large proportion of women in the sample. Psychopathy differs between men and women, and therefore the stability of the factor structure must be investigated separately for both genders what is not done. Due to the small sample size of men, I', afraid it's not possible to investigate the MI. This limitation should be addressed in the discussion, along with suggestions for future research.

Reply 1:

  • Thank you for pointing this out. We have included the following sentences in the manuscript in the Limitation section:

“A further and important limitation of the study is that no measurement invariance was considered. Kreis and Cooke already pointed out possible gender differences in 2011, but could not find any evidence for fundamental prototypical differences. The current study by Spohrmann, Mokros and Schneider [56] investigated the measurement invariance of different psychopathy models and instruments. They were mainly able to demonstrate higher mean scores in men compared to women. However, there were hardly any gender differences in the factor structure of the models/instruments investigated. In future studies on the CAPP-SR, more emphasis should be placed on measurement invariance by analyzing larger samples with a similar gender distribution.”

Comment 2:

  • The model fit is not good (with CFI values being low and RMSEA too high); modeling did not lead to significant improvements, possibly suggesting that the CAPP model might be too broadly conceptualized. This should be mentioned in the limitations section.

Reply 2:

Thank you for pointing this out. We have included the following sentence in the manuscript in the discussion:

“It is possible that the CAPP model is too broad, which often leads to difficulties when assessing model fit. However, conceptually, the broad conceptualization of psychopathy in the CAPP model is an advantage over alternative models because of its clinical benifit.

Comment 3:

  • Items were removed to achieve a better fit, but this can result in the loss of important information and potentially compromise the conceptual integrity of the original model. It's necessary to examine the removed items in relation to those that were included. What does a visual inspection of the removed items reveal? Are there theoretical arguments for why these items do not fit within the model? This must certainly be discussed, not only in the discussion section but also in the result section.

Reply 3:

Thank you for pointing out these extremely important points. However, in Chapter 3.3 (see page 7 to 8), the removal of items was discussed both statistically and in terms of content and placed in relation to previous studies and prototype analyses. Chapter 4 (Discussion) contains a more in-depth discussion and a comprehensive analysis of the effects of item removal on the model and the construct (see page 10 to11). For these reasons, we consider these comments to have already been included in the revision and have not included any further repetitions in the text.

However, we have added the following paragraph to the discussion to take the comment into account:

“Due to the removal of items or symptoms on the basis of statistical parameters (CFA), it should be noted that this procedure is logically opposed to the development of the CAPP, which was developed with the help of lexical and prototypical analyses. In principle, however, such a cross-check is also necessary to test the validity of a personality model. By removing items from the statistical analyses, significant information from the psychopathy construct could, of course, be lost. However, it should be noted that individual items or trait-descriptive adjectives were only removed at the lowest hierarchical level. This approach removes information from the model or the domains, but by no means fundamental information. Nor does it call into question the factor structure of the CAPP per se. Rather, future research on the CAPP should critically examine the postulated triangulation of three trait-descriptive adjectives (one item each), each of which depicts one symptom, and revise or better operationalize it if necessary.  “

Comment 4:

  • Additionally, also mention in the discussion that sample diversity is important and should be included in future research, along with the necessity of considering a broader range of demographic variables, such as age, educational level, and cultural background, in future studies.

Reply 4:

Thank you for pointing this out. We have included the following sentence in the manuscript in the discussion:

“Nevertheless, future studies should pay attention to a higher heterogeneity of the sample, especially with regard to socio-demographic variables (e.g. age, level of education, cultural background).”